# Stretchable Strain Sensor with Controllable Negative Resistance Sensitivity Coefficient Based on Patterned Carbon Nanotubes/Silicone Rubber Composites

**DOI:** 10.3390/mi12060716

**Published:** 2021-06-19

**Authors:** Rong Dong, Jianbing Xie

**Affiliations:** 1School of Mechatronic Engineering, Xi’an Technological University, Xi’an 710021, China; dongrong1981@163.com; 2Key Laboratory of Micro/Nano Systems for Aerospace, Ministry of Education, Northwestern Polytechnical University, Xi’an 710072, China

**Keywords:** strain sensor, negative resistance sensitivity coefficient, carbon nanotubes (CNTs)

## Abstract

In this paper, stretchable strain sensors with a controllable negative resistance sensitivity coefficient are firstly proposed. In order to realize the sensor with a negative resistance sensitivity coefficient, a stretchable stress sensor with sandwich structure is designed in this paper. Carbon nanotubes are added between two layers of silica gel. When the sensor is stretched, carbon nanotubes will be squeezed at the same time, so the sensor will show a resistance sensitivity coefficient that the resistance becomes smaller after stretching. First, nanomaterials are coated on soft elastomer, then a layer of silica gel is wrapped on the outside of the nanomaterials. In this way, similar to sandwich biscuits, a stretchable strain sensor with controllable negative resistance sensitivity coefficient has been obtained. Because the carbon nanotubes are wrapped between two layers of silica gel, when the silica gel is stretched, the carbon nanotubes will be squeezed longitudinally, which increases their density and resistance. Thus, a stretchable strain sensor with negative resistance sensitivity coefficient can be realized, and the resistivity can be controlled and adjusted from 12.7 Ω·m to 403.2 Ω·m. The sensor can be used for various tensile testing such as human motion monitoring, which can effectively expand the application range of conventional tensile strain sensor.

## 1. Introduction

Wearable technology tends to be more sophisticated than hand-held technology because it can provide skin-mountable biofeedback and tracking of physiological function. In particular, stretchable and wearable strain sensors are needed for several potential applications including human motion and health detection, human-robot interaction, artificial skin, smart clothing, and so forth [1,2].

Stretchability and sensitivity are the key features of strain sensor which can be described by Strain (ε) and Gauge Factor (GF), respectively. In recent years, several types of stretchable strain sensors have been proposed by using nanomaterials coupled with flexible and stretchable elastomer. To improve stretchability, the conductive particles such as nanoparticles (NPs) [3,4,5,6], carbon nanotubes (CNTs) [7,8,9,10], silver nanowires (AgNWs) [11,12], and graphene [13,14] are typically coated on soft elastomer, such as PDMS [13], Ecoflex [12,15], silicone elastomer [16,17], rubber [18,19], dragon-skin elastomer [20] et al. In [15], super-stretchable, skin-mountable, and ultra-soft strain sensors are presented by using carbon nanotube percolation network-silicone rubber nanocomposite thin films, the stretchability can achieve 500%, but the GF is only 1–2.5. In [21], AuNWs–latex rubber nanocomposite are used to achieve a new type of sensor featured a GF of ≈ 9.9 and stretchability of >350%. In [22], a strain sensor with Ecoflex rubber elastic substrate and rGO/DI sensing liquids is designed to make the super-elasticity possible. The stretchability can achieve 400%, and the GF is 31.6.

It can be found from the above research status, the resistance sensitivity is all positive, so in order to extend the application range of the stretchable strain sensor, this paper presents a controllable negative resistance sensitivity coefficient stretchable strain sensor. This is the first time that a negative resistance sensitivity coefficient stretchable strain sensor is proposed.

In [23], a strain sensor based on the sandwich-like PDMS/CNTs/PDMS composite is proposed. The strain sensor not only presents a good optical transmittance, but also gains the ability to monitor both the subtle motions of facial expressions and the large motions of human joints. On this basis, we found that the sandwich-like structure can also obtain a negative resistance sensitivity coefficient, which can further expand the application range of the sensor and its specific application will be developed in the following research.

## 2. Principle and Design

The typical structural composition of stretchable strain sensors is shown in Figure 1, nanomaterials are coated on soft elastomer, the elastomer core is a rectangular structure, the length (L), width (W), and thickness (T) are shown in Figure 1a,b. A “dragon-skin” silica gel with maximum strain up to 900% is used for soft elastomer, and the nanomaterials are single-walled carbon nanotubes. At this point, the resistance of the sensor depends on resistivity of the powder material, particle size of powder, and especially compactness of the powder material. As a stretchable strain sensor, when the elastomer core is stretched, as shown in Figure 1c, part of the transverse connection in the powder material will be broken, and the compactness of the powder material is reduced, thereby causing the sensor resistance to increase. Therefore, the stretchable strain sensor prepared by this structure scheme only has a positive resistance sensitivity coefficient. 

In order to extend the application range of the stretchable strain sensor, we present a controllable negative resistance sensitivity coefficient stretchable strain sensor, as shown in Figure 2. First, nanomaterials are coated on the soft elastomer. Then, a layer of silica gel is wrapped on the outside of the nanomaterials, as shown in Figure 2a. The thickness of the elastomer and nanomaterials is T_ec_ and T_cnts_, respectively. When the elastomer is stretched, as shown in Figure 2b, although the lateral compactness of powder material is reduced, the longitudinal compactness increases. 

Lateral compactness denoted the tightness of CNTs along the tensile direction. With the increase of tensile length, part of CNTs was disconnected, leading to the decrease of lateral compactness. Longitudinal compactness represents the compactness in the direction of the thickness of CNTs. With the increase of tensile length, CNTs have more contact under the longitudinal extrusion effect, which leads to the increase of longitudinal compactness.

Therefore, it is expected to obtain a stretchable strain sensor with a negative resistance sensitivity coefficient through reasonable pattern design.

Normally, the powder resistivity measured by the pressurization method can be expressed as:(1)$$\rho_{0}=\frac{V}{I}\frac{A}{h}$$
where *ρ*_0_ is the initial resistivity of powder resistor, *V* is the applied voltage, *I* is the measured current, *h* is the powder thickness along the current direction, *A* is the cross-sectional area. It should be noted that powder resistivity varies a lot with powder porosity, humidity, or temperature.

Further, the resistance (*R*) of the stretchable strain sensor can be expressed as:(2)$$R=\rho \frac{L}{A}$$
where *ρ* is the resistivity of the sensor, *L* length of sensor. For the stretchable strain sensors designed in this paper, its resistivity, length, and cross-sectional area will change when stretched.

In order to verify the influence of different pattern structures on resistance, we designed several embedded resistance schemes as shown in Figure 3. Three stretchable strain sensors with different nanomaterial patterns are designed to verify the feasibility of the negative resistance sensitivity coefficient, which are defined as Type A, Type B, and Type C, respectively, as shown in Figure 3a–c, at the same time, typical stretchable strain sensors with the same patterns of Type A are designed as a comparison, which are defined as Type D, as shows in Figure 3d. The cross section of all resistors is rectangular. Since the unencapsulated structures corresponding to Type B and Type C will deform during stretch, it is impossible to compare the tensile effect. Therefore, this paper only carried out comparative experiments through Type A.

## 3. Fabrication

The fabrication process of the designed negative resistance sensitivity coefficient stretchable strain sensor is shown in Figure 4. The elastomer core is made of a kind of “dragon-skin” silica gel which has a maximum strain up to 900%, so it can provide enough deformation for the designed stretchable strain sensor.

First, insert two copper electrodes into the configured silica gel and then cure at room temperature (25 °C) for 1 h, as shown in Figure 4a. At this time, the surface of the silica gel has a good viscosity, but it has been substantially formed and can be cut into a desired shape, as shown in Figure 4b. The carbon nanotube powder is evenly dispersed on the surface of the silica gel elastomer core and compacted to obtain the stretchable strain sensor. At this time, after the further curing process, the typical structural of stretchable strain sensors can be obtained, as shown in Figure 4c. 

At this stage of the process, we have the stretchable strain sensor of Type D, as shown in Figure 5a. This is a stretchable strain sensor with a positive resistance sensitivity.

However, the negative resistance sensitivity coefficient stretchable strain sensor proposed in this paper is re-encapsulated on this basis, as shown in Figure 4d. The overall structure shown in Figure 4c is encapsulated using the same “dragon-skin” silica gel, after that, the negative resistance sensitivity coefficient stretchable strain sensor can be obtained by curing at room temperature (25 °C) for 2 h.

The fabricated stretchable strain sensor (Type D) is shown in Figure 5. The thickness of the carbon nanotube layer is about 80 μm.

The fabricated stretchable strain sensors (Type A, Type B, and Type C) are shown in Figure 6. It can be seen that CNTs are well wrapped in silica gel and can be stretched together with silica gel.

## 4. Characterization and Discussion

In order to verify the feasibility of the stretchable strain sensor with negative resistance sensitivity coefficient, a comparative test is carried out in this paper. The stretchable strain sensor is stretched to different lengths and its resistance is recorded. Figure 7 shows the stretch test of the fabricated stretchable strain sensor.

The initial length of the sensor is 25 mm (Figure 7a), and it was stretched to 50 mm (Figure 7b), 75 mm (Figure 7c), and 90 mm (Figure 7d), respectively, to verify the resistance performance after stretching. It can also be seen from Figure 6 that the width of the resistor is basically unchanged, but the thickness gradually decreases with the stretching.

The resistance value and its change curve of the stretchable strain sensors after stretching are shown in Table 1. For the stretchable strain sensors with Type A, Type B, and Type C, as the stretched length of the sensor increases, the resistance increases continuously. However, for Type D, the resistivity increases rapidly with the increase of the stretch.

Figure 8 shows the SEM view of the fabricated stretchable strain sensors, before stretching, the thickness of the carbon nanotube layer is about 32 μm, as shown in Figure 8a, and after stretching the thickness decreased to 17.2 μm (ε = 100%) and 10.6 μm (ε = 200%), as shown in Figure 8b,c. This clearly shows that the thickness of the sensor is compressed as it is stretched, therefore, the change in resistance is no longer monotonically increasing.

According to the above experimental results, the resistivity of the sensor can be obtained, as shown in Table 2. It can be seen that for the stretchable strain sensors with Type A, Type B, and Type C, as the stretched length of the sensor increases, the resistivity increases continuously. This means that during the stretching process, the squeezing effect of the elastomer on the nanomaterial is greater than the stretching effect. When there is no external elastomer wrap (Type D), the resistivity increases rapidly with the increase of the stretch.

We can also see from Table 2 that the initial resistance between Type A and Type D is a lot different. This is because the liquid silica gel affected the contact of part of the carbon nanotubes when the Type D structure was repackaged into the Type A structure, leading to the decline of its initial resistance.

So, we can get the relationship curve between sensor length and resistivity, as shown in Figure 9. It can be seen that the stretchable strain sensors with Type A, Type B, and Type C have a negative resistance sensitivity coefficient.

As shown in Figure 9, the resistance sensitivity coefficient of Type D within ε = 100% is 12.23, by contrast, the resistance sensitivity coefficient of Type A within ε = 100% is negative 12.07 Ω·m/mm. At the same time, the resistance sensitivity coefficient can be controlled according to different lengths and patterns.

Therefore, the stretchable strain sensor with negative resistance sensitivity designed in this paper can expand the application of this device in many fields such as human motion monitoring.

## 5. Conclusions

This paper presents the design, fabrication of a novel stretchable strain sensors with a controllable negative resistance sensitivity coefficient. A stretchable stress sensor with silica gel/CNTs/silica gel sandwich structure is designed to realize the negative resistance sensitivity coefficient. The stretchability of the fabricated stretchable strain sensors can achieve up to 260%, resistance sensitivity coefficient is negative at 12.07 Ω·m/mm, and the adjustable control of resistivity from 354.3 Ω·m to 13.2 Ω·m, which can greatly expand the application field of the stretchable strain sensors such as human motion monitoring.

## Figures and Tables

**Figure 1 micromachines-12-00716-f001:**
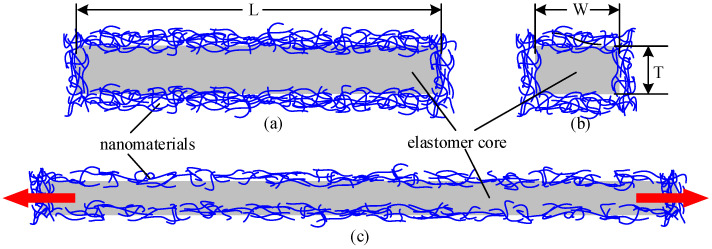
Typical structural composition of stretchable strain sensors, front view (**a**) and section view (**b**) before stretch and front view after stretch (**c**).

**Figure 2 micromachines-12-00716-f002:**
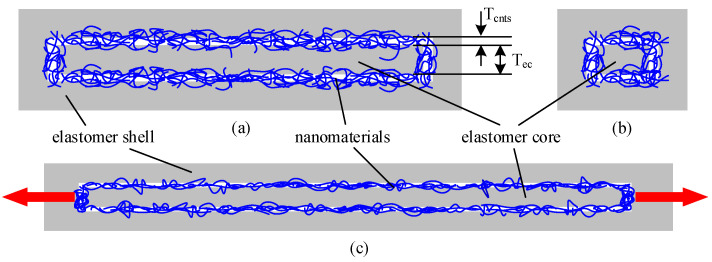
Controllable negative resistance sensitivity coefficient stretchable strain sensor, front view (**a**) and section view (**b**) before stretch and front view after stretch (**c**).

**Figure 3 micromachines-12-00716-f003:**
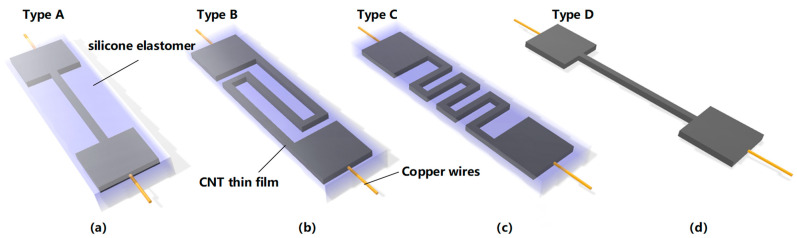
Designed controllable negative resistance sensitivity coefficient stretchable strain sensor (**a**–**c**) and typical structural of stretchable strain sensor (**d**).

**Figure 4 micromachines-12-00716-f004:**
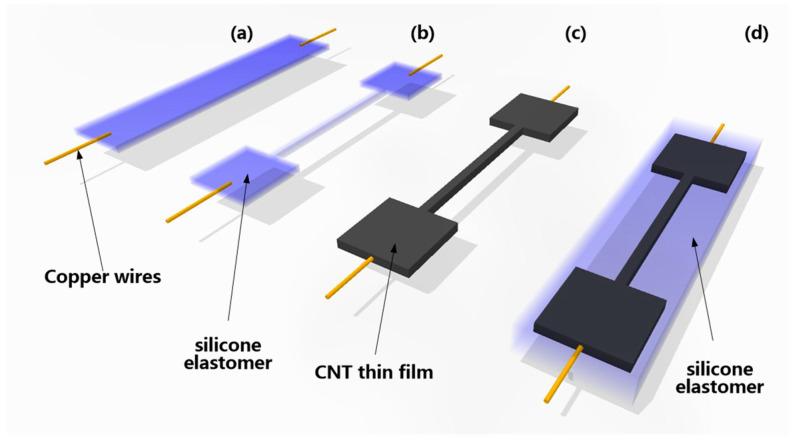
The fabrication process of the designed negative resistance sensitivity coefficient stretchable strain sensor. (**a**) Preparation of elastomer core and copper electrodes. (**b**) Silicone elastomer forming. (**c**) Preparation of CNTs film. (**d**) Silica gel secondary package.

**Figure 5 micromachines-12-00716-f005:**
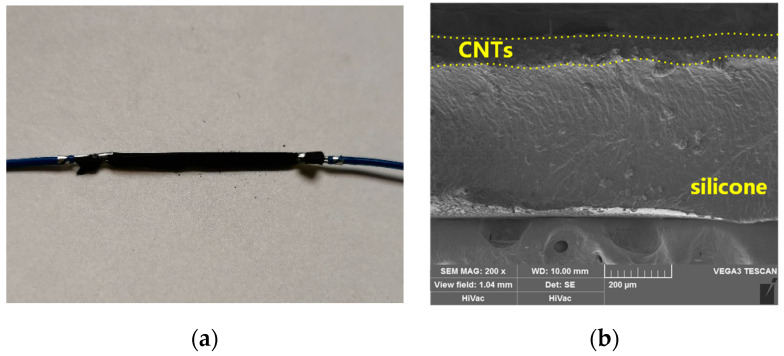
The fabricated stretchable strain sensor (**a**) and the profile SEM view (**b**).

**Figure 6 micromachines-12-00716-f006:**
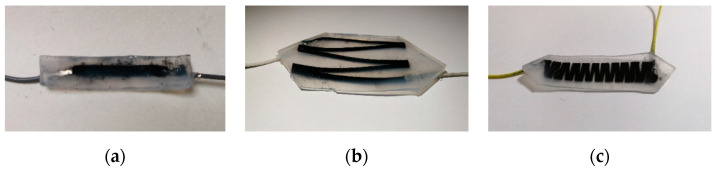
The fabricated stretchable strain sensor (**a**) Type A, (**b**) Type B, (**c**) Type C.

**Figure 7 micromachines-12-00716-f007:**
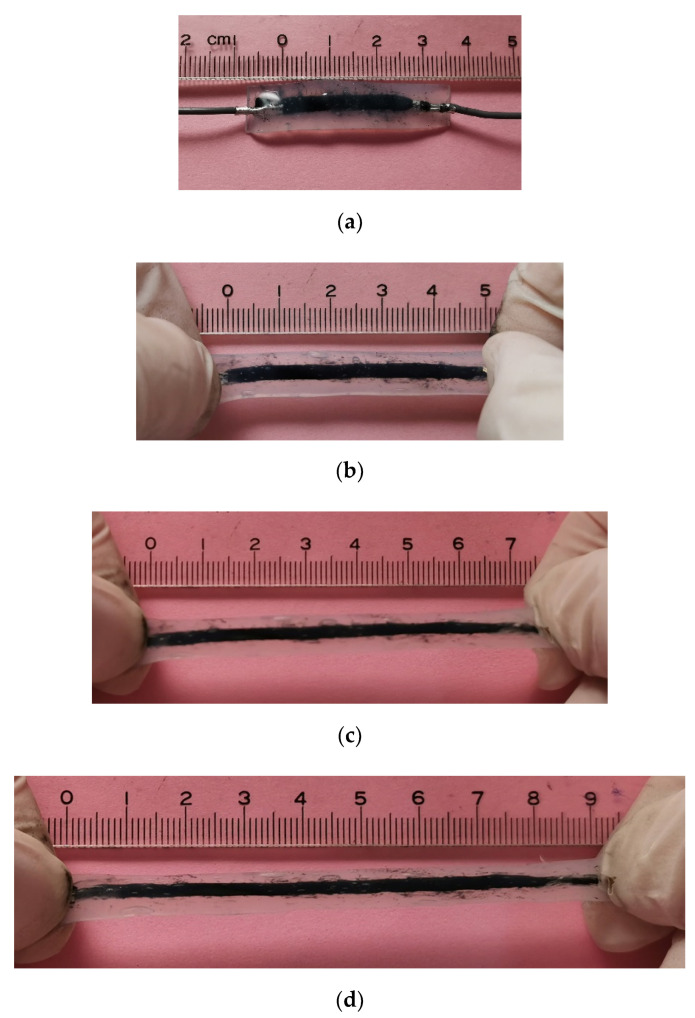
Stretch test of the fabricated stretchable strain sensor at ε = 0% (**a**), ε = 100% (**b**), ε = 200% (**c**), and ε = 260% (**d**).

**Figure 8 micromachines-12-00716-f008:**
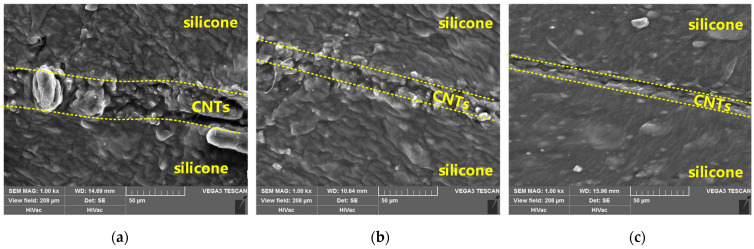
SEM view of the fabricated stretchable strain sensor at ε = 0% (**a**), ε = 100% (**b**) and ε = 200% (**c**).

**Figure 9 micromachines-12-00716-f009:**
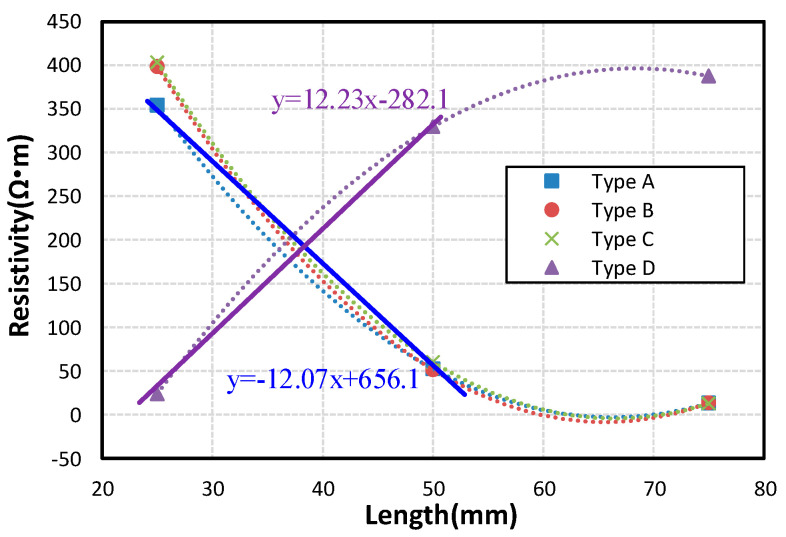
The relationship curve between sensor length and resistivity.

**Table 1 micromachines-12-00716-t001:** Resistance test result of the stretchable strain sensors (The average of five measurements) @ 25 °C.

Sensor Type	Resistance @ ε = 0%(MΩ)	Resistance @ ε = 50%(MΩ)	Resistance @ ε = 100%(MΩ)	Resistance @ ε = 200%(MΩ)	Resistance @ ε = 260%(MΩ)
A	34.6 ± 12.4	27.4 ± 7.5	21.5 ± 6.9	14.2 ± 4.6	4.9 ± 1.8
B	194.5 ± 28.7	142.7 ± 18.7	105.6 ± 10.7	72.7 ± 8.7	36.8 ± 9.4
C	151.2 ± 20.4	120.4 ± 14.5	94.7 ± 10.2	52.4 ± 6.4	24.3 ± 7.5
D	1.35 ± 0.86	16.4 ± 4.7	42.2 ± 7.5	80.1 ± 10.1	126.2 ± 15.6

**Table 2 micromachines-12-00716-t002:** Resistivity of the stretchable strain sensors @ 25 °C.

/	Type A			Type B			Type C			Type D		
Length (μm)	25	50	75	125	250	375	96	192	288	25	50	75
Resistance (MΩ)	34.6	21.5	14.2	194.5	105.6	72.7	151.2	94.7	52.4	1.35	42.2	80.1
Tec (mm)	1	0.55	0.3	8	7.1	6.6	8	7.1	6.6	8	7.1	6.6
Tcnts (μm)	32	17.2	10.6	32	17.2	10.6	32	17.2	10.6	55	55	55
Resistivity (Ω·m)	354.3	52.5	13.2	398.3	51.5	13.5	403.2	60.2	12.7	23.7	329.5	387.6

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
