# Peer review of "Stretchable Strain Sensor with Controllable Negative Resistance Sensitivity Coefficient Based on Patterned Carbon Nanotubes/Silicone Rubber Composites"

_micromachines, 2021, doi:10.3390/mi12060716_

Round 1

Reviewer 1 Report

This manuscript presents the fabrication of stretchable strain sensors with negative resistance sensitivity. The resistivity of these sensors can be controlled and adjusted. This manuscript must be improved considering the following comments:

1.-Abstract section must include more information about the design and materials of the sensors. In addition, this section should add a conclusion or application of the proposed sensors.

2.-Introduction section must include more recent information of stretchable sensors reported in the literature. This section must add the main advantages of the proposed sensors in comparison with other reported in the literature. Which are the limitations or challenges of the proposed sensors?

3.-Section of principle and design. This section is short. Authors should improve this section including more information about the design, dimensions, limitations and materials of the different proposed sensors.

4.-Fabrication section requires more detail information about the different stages of the fabrication process of the proposed sensors.

5.- Which is the influence of the temperature on the behavior of the sensors?

6.- Which are the main limitations or challenges of the proposed sensors?

7.-Which is the influence of the CNTs on the behavior of the proposed sensors?

8.-Authors should include more measurements of the behavior of the proposed sensors.

9.-The fourth section  requires more discussion of the main results.

10.-Conclusion section must include more information about the results, main advantages and potential applications of the proposed sensors.

Reviewer 2 Report

The manuscript, "Stretchable Strain Sensor with Controllable Negative Resistance Sensitivity Coefficient based on Patterned Carbon Nanotubes/Silicone Rubber Composites" by Rong and Jianbing, describes an interesting structure with the negative resistivity coefficient with elastomer shell. By reading through the manuscript, a further elaboration of why this structure can provide a negative resistivity and a more detailed experimental description is needed to help the reader better understand the design and phenoma. The manuscript could be published after the updates on the details.

  1. The author designed 6 type of sensors but only include the results for (a)-(d). What are the results of (e) and (f)?
  2. How many samples have the author tested for each sensor design? It is intereting to see how the resistance varies and how resistance coefficient changes across different samples. Please also add error bars for all the plots.
  3. I noticed that the initial resistance between (a) and (e) are a lot different. Why? How about (b) and (f), (c) and (g)? If the shape is the same, why the resistance are very different?
  4. With the new structure, are there any chances to realize positive resistance as well?
  5. Line 59-67, please add more explanation about what is "lateral compactness" and what is "longitudinal compactness". How each of the compactness influence the parameters in equ (1) and (2).
  6. Please add more info on the fabrication part, include material, equipment info, process time and temperature, etc.
  7. Please provide more info about how negative resistitivy can be used/beneficial for real application. Some elaborations or extra references is recommended.

Round 2

Reviewer 1 Report

This version of manuscript has been improved considering the reviewer's comments.

Reviewer 2 Report

The author has addressed all my comments and concerns. The manuscript is recommended for publication.